# HFNet-SLAM: An Accurate and Real-Time Monocular SLAM System with Deep Features

**DOI:** 10.3390/s23042113

**Published:** 2023-02-13

**Authors:** Liming Liu, Jonathan M. Aitken

**Affiliations:** Department of Automatic Control and Systems Engineering, The University of Sheffield, Sheffield S10 2TN, UK

**Keywords:** simultaneous location and mapping, deep features, monocular localisation

## Abstract

Image tracking and retrieval strategies are of vital importance in visual Simultaneous Localization and Mapping (SLAM) systems. For most state-of-the-art systems, hand-crafted features and bag-of-words (BoW) algorithms are the common solutions. Recent research reports the vulnerability of these traditional algorithms in complex environments. To replace these methods, this work proposes HFNet-SLAM, an accurate and real-time monocular SLAM system built on the ORB-SLAM3 framework incorporated with deep convolutional neural networks (CNNs). This work provides a pipeline of feature extraction, keypoint matching, and loop detection fully based on features from CNNs. The performance of this system has been validated on public datasets against other state-of-the-art algorithms. The results reveal that the HFNet-SLAM achieves the lowest errors among systems available in the literature. Notably, the HFNet-SLAM obtains an average accuracy of 2.8 cm in EuRoC dataset in pure visual configuration. Besides, it doubles the accuracy in medium and large environments in TUM-VI dataset compared with ORB-SLAM3. Furthermore, with the optimisation of TensorRT technology, the entire system can run in real-time at 50 FPS.

## 1. Introduction

Simultaneous localization and mapping (SLAM) techniques are the foundation of robot autonomy within the wider world by enabling both map building and determining the location of a robot in a hitherto unknown environment. During the last decade, with the increasing popularity of low-cost cameras, significant development has taken place in visual-based SLAM systems.

Among visual SLAM systems, feature-based methods have produced outstanding systems with increasing robustness and accuracy, such as ORB-SLAM3 [1], VINS-Mono [2], and OKVIS [3]. Most of these systems heavily rely on two types of image features based on hand-crafted principles: local and global features. However, these features may be corrupted and affect the performance of SLAM systems by challenging factors in the real world, including sparse texture, poor image quality, and occlusion [4,5,6].

Local features include the points of interest called keypoints and their descriptors. Extracting and matching these features in several images can provide pose estimation information [7]. Recent development of deep convolutional neural networks (CNNs) has demonstrated strong ability in feature extraction [8]. Experiments show local features based on deep CNNs have higher repeatability and matching scores even in complex environments [9], which is superior to hand-crafted features [10]. By replacing hand-crafted features with deep ones, SLAM systems can obtain more reliable environmental information [11].

Global features, also referred to as global descriptors, provide compact representations for images. Global features are essential for loop detection in SLAM systems because they help robots identify previously visited places to mitigate the accumulated drift and build a consistent map. The bag-of-word (BoW) [12] method is the most widely used method for extracting global features from images. However, many researchers state that CNN-based loop detection methods outperform the BoW method and provide more robustness to the variance of illumination, moving objectives, and seasonal changes [13,14,15].

To implement the practicability of binding both local and global deep features with a real-time SLAM system and discuss the impact, this work proposes an accurate and robust monocular visual-inertial SLAM system called HFNet-SLAM. This system is a combination and extension of the well-known ORB-SLAM3 SLAM framework [1] and a unified CNN model called HF-Net [16]. To ensure real-time performance, TensorRT technology is used to optimise the CNN model. After implementation, the performance of HFNet-SLAM has been well validated in various environments among the EuRoC and TUM-VI datasets to illustrate the accuracy and efficiency of the whole system.

## 2. Background Literature

### 2.1. Visual SLAM

There is extensive research on visual SLAM systems. Davison proposed MonoSLAM in 2003, beginning the era of monocular SLAM [17]. This algorithm uses an extended Kalman filter (EKF) to achieve real-time localization with sparse prior scene knowledge. However, this pure visual SLAM performs poorly with occlusions, motion blur, and trackable texture sparsity. To address this problem, the multi-state constraint Kalman filter (MSCKF) [18] allows the incorporation of an inertial measurement unit (IMU) with vision. MSCKF is a tightly coupled visual-inertial algorithm in which several camera poses and IMU measurements are updated and jointly optimized.

Although EKF-based SLAM systems are simple and straightforward, researchers are always striving to improve their accuracy. Keyframe-based approaches and bundle adjustment (BA) optimization [19] are used to improve the accuracy at the expense of higher computation. VINS-Mono [2] is a very robust odometry estimation system. It maintains the recent states of map points and cameras in a fixed-size sliding window and updates them by iteratively solving BA. ORB-SLAM3 [1] strengthens the ability to reuse a map to build a co-visibility graph. It uses short-term, mid-term, and long-term data association in local visual-inertial BA and an effective IMU initialization technique to achieve remarkable accuracy in indoor environments.

The rapid development of neural networks brings considerable benefits to visual SLAM technology. Li et al. proposed a deep convolutional neural network (CNN) to accurately estimate depth information for mapping from two consecutive monocular images [20]. DeepVO [21] introduced an end-to-end recurrent CNN (RCNN) for pose estimation. This algorithm directly inferred poses from an RGB-D stream without the pipeline of traditional visual odometry (VO), like feature matching, pose optimisation, and so on. DeepSLAM [22] extended the previous ideals and employed three different types of CNNs for tracking, mapping, and loop closing. RDS-SLAM [23] incorporated a deep learning application for semantic segmentation into a SLAM framework to reduce the negative impact of moving objects on accuracy. Although some deep SLAM work has achieved impressive accuracy improvements, real-time performance and the ability to transform the network for various scenarios remain challenges [24].

### 2.2. Visual SLAM with Deep Features

#### 2.2.1. Local Features

Most of the current leading visual SLAM systems rely on matching 2-D keypoints in frames or 3-D map points according to the similarity of their descriptors. Hand-crafted features, including ORB [25], Shi-Tomasi [26], and BRISK [27], have been widely used for their efficiency [7]. These traditional features, however, have been shown to be inaccurate and unstable, reducing the robustness of tracking and mapping in classical localization methods [28,29,30]. Local features emerging from deep learning significantly outperform these traditional features. DeTone et al. introduced a self-supervised fully convolutional model called SuperPoint to compute pixel-wise features with outstanding matching ability [9]. Ono et al. designed a deep network that allows training without hand-crafted priors and produces robust features in both indoor and outdoor environments [31]. SLAM systems can get such benefits by applying deep features. The GCN-SLAM proposed the GCNv2 network to replace the ORB algorithm for extracting local features in the ORB-SLAM2 framework. The authors concluded that the GCNv2 net produced better distributed features than the hand-crafted features, which led to a better quality of tracking [32]. In addition, DF-SLAM also successfully combined learned features with traditional visual SLAM systems, demonstrating good accuracy and real-time performance [33].

#### 2.2.2. Global Features

Global features provide an effective solution for image retrieval and place matching. The majority of modern SLAM systems rely on BoW [12] for place recognition. This method clusters local features into visual words and aggregates the presence or absence of visual words to generate the representation of an image [34]. However, significant drawbacks have been reported in recent years. Garcia-Fidalgo et al. stated that the pre-trained vocabulary in an offline step only achieves correct results in generic environments and may lead to false detection in unexplored environments [35]. Zhang et al. highlighted that the BoW descriptors merely represented the appearance of features without any geometric relationship, which thereby led to poor performance in places with similar features [36]. However, deep CNN-based approaches have shown great power in representing images. Arandjelovic et al. proposed NetVLAD, which aggregated high-level features in convolution layers to produce a global feature in a fixed-length vector [37]. Hausler et al. further extended NetVLAD to include a multi-scale fusion of features to enhance place recognition performance in challenging environments [38]. Such efforts have produced remarkable results in a variety of SLAM algorithms. Yang et al. introduced the NetVLAD model into a multi-camera visual SLAM system, and they declared that loop detection based on deep learning methods showed excellent robustness to some factors, such as rough roads, different viewpoints, and changing illumination in off-road scenarios [39]. Kuse et al. designed a loop detection model based on deep features. Their experiments showed that deep global features provided a more comprehensive description of images compared with traditional methods, leading to high recall and precision in loop detection [40].

#### 2.2.3. Joint Local and Global Features

Although the deep features bring significant improvements, inferring multiple models for extracting local and global features can be extremely expensive for a real-time SLAM system. To solve this problem, it is natural to consider combining two types of features in a single model. Cao et al. devised a model called DELG, which unified local and global features into a single deep model to reduce the computation cost by sharing calculations [41]. Based on the same ideal, the authors of [16] used MobileNet [42] as the backbone, combining a SuperPoint encoder [9] and a NetVLAD layer [37] to output local and global features. Furthermore, they introduced a distillation training method to reduce the complexity of the model. Their model, called HF-Net, can run in real time and has outstanding performance in complex environments. Although these unified models have achieved success in the structure from motion (SFM) problem, they have not been widely accepted in the SLAM domain. DXSLAM [43] is the first step towards building such a system. It incorporated HF-Net into the framework of traditional RGB-D SLAM and proved that the introduction of unified models can improve the performance of SLAM systems with RGB-D sensors. However, the disadvantages of DXSLAM are apparent. Firstly, DXSLAM does not use the image pyramid strategy while extracting local features, producing a small number of keypoints, and corrupting the accuracy of the system in a complex environment. Secondly, this system failed to take full advantage of global features generated by HF-Net. It still relies on the BoW method for loop detection and keypoint matching. Last but not least, the evaluation results of DXSLAM are unconvincing because the authors only demonstrate the quantitative error comparison in five sequences in similar environments.

HFNet-SLAM is based on the work mentioned above. In this work, the ORB-SLAM3 framework is used as the structural backbone. HF-Net optimised by TensorRT is chosen for feature extraction because of its special design for efficiency, though its production may not be the most accurate among other similar works. To take full advantage of the deep features, HFNet-SLAM proposes effective and accurate keypoint processing and loop detection modules without the assistance of the hand-crafted features and BoW algorithm. HFNet-SLAM, in contrast to most similar works that use RGB-D sensors, is intended for monocular cameras. The work of HFNet-SLAM is more fundamental and can be further extended to stereo and RGB-D SLAM.

## 3. HFNet-SLAM

### 3.1. Overview

Figure 1 shows the main components of the proposed monocular visual-inertial SLAM system, called HFNet-SLAM. This system is built on ORB-SLAM3 [1] and thus shares plenty of the same components with it. The system starts with a tracking thread, in which the local features are tracked and IMU data are integrated to estimate the pose of the current frame. Once a frame is decided to be a keyframe by the tracking thread, further operations, including global feature extraction, keypoint triangulation, and local BA, are performed on the keyframe in the mapping thread. Based on the global features, an effective loop closure method is proposed to eliminate the accumulated drift in the loop closure thread.

Compared with ORB-SLAM3, the distinguishing feature of HFNet-SLAM is that the local features (keypoints and corresponding descriptors) and global features are extracted from a deep CNN, HF-Net, instead of the traditional ORB and BoW methods. Therefore, some parts of the system, including feature extraction, keypoint matching, and loop detection, are customized for this modification.

### 3.2. Feature Extraction

The deep convolutional neural network, HF-Net [16], is used to extract local and global features from the input frames. There are two potential improvements to increase the adaptability of HF-Net in SLAM systems. Firstly, the original HF-Net model does not support the widely used image pyramid strategy in SLAM systems. Besides, the model calculates global features for every frame, which is not required by SLAM systems, causing unnecessary calculation. Therefore, the original HF-Net is divided into two parts: the local and global parts, each running on two threads, respectively. The process of feature extraction is plotted in Figure 2.

The local part of HF-Net contains a MobileNet [42] and a SuperPoint [9] encoder. It takes an image as input and generates keypoints and corresponding descriptors, as well as the shared features for further calculation. Several local parts are deployed on the tracking thread. In this thread, the input frame is copied and iteratively scaled down to form an image pyramid. For each level in the pyramid, the scaled image passes through a local part of the model to collect the local features.

The global part of HF-Net, consisting of another MobileNet and a NetVLAD layer [37], is employed in the mapping thread for processing keyframes. The previously calculated shared features from the tracking thread are reused to obtain the global features for the input image.

This process has several benefits. Firstly, the image pyramid enables multi-detection for the same image with different resolutions, which increases the number and scale-invariant characteristics of local features. Besides, the global features are only further calculated for keyframes. The reuse of shared features and the avoidance of unnecessary calculations effectively reduce the cost of the feature extraction process.

The original HF-Net model is implemented in TensorFlow. The performance of model prediction can be improved by the TensorRT toolkit [44]. To reduce the latency of inference, the pre-trained HF-Net model is optimized and deployed with the TensorRT engine on NVIDIA GPU devices.

### 3.3. Keypoint Matching

Keypoint matching strategies with good performances are essential for HFNet-SLAM. These strategies find the correspondences between keypoints in frames and maps to provide geometric constraints for pose estimation.

HFNet-SLAM has two types of matching strategies for different scenarios. 3-D–2-D matchers are used for tracking 3-D points on maps with 2-D keypoints on frames. Matchers project map points onto the image and only match keypoints nearby the projection area.

However, 2-D–2-D matchers for tracking 2-D keypoints in different frames can be extremely expensive since matchers have to match a large number of possible pairs. To bound the computation complexity, traditional systems, including ORB-SLAM3 and DXSLAM, only match a limited number of putative matches between keypoints provided by a pre-trained BoW vocabulary tree [1,43]. In order to get rid of the dependence on the BoW algorithm, HFNet-SLAM has to use brute-force matchers on the full set of possible keypoint pairs. This unacceptable computation cost may corrupt the real-time performance.

The process of matching can benefit from vectorization technology. The descriptor of a keypoint generated by HF-Net is a vector of 256 floating numbers, denoting as $v\in R^{256}$. Considering an image with a set number of keypoints, their descriptors can be concatenated to form a matrix $V={\left[v^{1},v^{2},\cdots ,v^{n}\right]}^{T}\in R^{n\times 256}$, where *n* is the number of keypoints. HFNet-SLAM uses $L_{2}$-norm between descriptors to measure the matching score. Since the descriptor matrix is normalized, the pairwise matching score between rows of $V_{1}$ and $V_{2}$ can be simplified as:(1)$$score=\sqrt{V_{1}V_{1}^{T}+V_{2}V_{2}^{T}-2V_{1}V_{2}^{T}}\approx \sqrt{2-2V_{1}V_{2}^{T}}$$

The matrix multiplication $V_{1}V_{2}^{T}$ can be sped up by using parallel calculation and single-instruction-multiple-data (SIMD) technology on modern CPUs. Although this method cannot avoid unnecessary calculations, it greatly improves the CPU throughput to increase computational efficiency [45].

### 3.4. Loop Detection

Loop detection is essential for the SLAM system to eliminate the accumulated errors. Using the BoW method for loop detection is the most popular choice for recently proposed SLAM systems. However, this method has limited power to describe the whole image [36] and only obtains 50%ߝ80% precision and recall [12]. As a result, the traditional BoW-based loop detection method tends to employ stringent verification criteria to avoid false positives that could harm the SLAM system. As Figure 3 shows, in ORB-SLAM3, loop candidates are selected through multiple factors. Such verification can improve the precision of the retrieval at the cost of reducing recall and efficiency.

Due to the high accuracy of CNN-based global features, the proposed loop detection in HFNet-SLAM is straightforward and effective. The process begins with an image retrieval module, as shown in Figure 3. This module calculates the similarity between images according to their global features. Several candidate images from the database are selected based on a high score with the query image. The similarity score between two global features *x* and *y* is computed by $L_{2}$-norm:(2)$$s\left(x,y\right)=1-\sqrt{\sum_{i=1}^{n}{\left(x_{i}-y_{i}\right)}^{2}}$$
where, *x* and *y* are the global feature vectors from two images. n=4096 is the dimension of vectors. s(x,y) is the similarity score of two images, and the higher the similarity, the greater the possibility of becoming a loop.

After several loop candidates are detected by image retrieval, geometric verification is used to improve the accuracy of loop closure. First, keypoint matchers are used to establish feature correspondences between the candidate frame and the query frame. For these matches, a 3-D alignment with RANSAC [46] is used for outlier rejection and transformation estimation. After geometric verification on three co-visible keyframes, this candidate frame is regarded as a genuine loop.

This deep learning-based loop detection is both accurate and effective. This improvement leads to the better performance achieved by HFNet-SLAM in contrast to ORB-SLAM3 in the TUM-VI and runtime performance experiments.

## 4. Results

In this section, the HFNet-SLAM system is implemented in C++, and the HF-Net model is calculated with the TensorRT C++ API for a uniform programming language and better efficiency. After implementation, the entire system is evaluated in public benchmarks against different state-of-the-art systems. Two different experiments, including SLAM experiments and runtime evaluation, are performed to evaluate the performance in terms of accuracy, stability, and computational cost.

During evaluation, the experiments of HFNet-SLAM run on an Intel Core i7-10750H CPU, at 2.6 GHz, with 16 GB memory and an NVIDIA RTX 2070 with Max-Q Design GPU, with 8 GB memory. Considering the various environmental features in test benchmarks, the number of extraction features for every input image is adjusted to get a trade-off between reducing computational cost and collecting sufficient information in a complex environment. Besides, other systems available in the literature run with their default configuration.

### 4.1. SLAM Experiment

In this experiment, the HFNet-SLAM and other state-of-the-art systems are performed on the EuRoC [47] and TUM-VI [48] datasets to validate the overall accuracy and robustness. For each sequence in the datasets, the poses for all frames are evaluated by different SLAM algorithms. Then, the estimated poses are aligned with the corresponding ground-truth positions using Sim(3) transformation in the pure monocular case, and using SE(3) transformation in the visual-inertial case. After that, the rms ATE (absolute trajectory error) [49] is calculated as the evaluation metric to compare the performance of various systems. For the visual-inertial case, the scale errors are also calculated using s from Sim(3) alignment.

#### 4.1.1. Monocular SLAM on EuRoC Dataset

The EuRoC dataset is a widely used and accepted benchmark by SLAM developers. This dataset is composed of 11 sequences of small indoor environments. It provides stereo images recorded at 20 FPS, IMU measurements at 200 Hz, as well as full ground-truth data of camera motions [47]. During evaluation, only the images from the left camera are used for the monocular SLAM. HFNet-SLAM takes 752 × 480 images as inputs and extracts 675 keypoints for every image.

Figure 4 demonstrates the evaluation results of the HFNet-SLAM with monocular and monocular-inertial configurations. The obtained results are also compared with other relative state-of-the-art SLAM systems, including ORB-SLAM3 [1], DSM [50], and VINS-Mono [2].

In the monocular-inertial case, HFNet-SLAM has the most precise results among these systems, obtaining errors below 5 cm for most of the sequences. Compared with ORB-SLAM3, HFNet-SLAM can provide a more accurate scale evaluation, leading to a slight improvement in the average ATE. In this case, the estimated results by different algorithms in *V202* sequence are plotted in Figure 5a. As can be seen, plenty of regions are revisited during the tracking. HFNet-SLAM and ORB-SLAM3 use mid-term data association to reuse previous regions, leading to low trajectory errors of 2 cm, while VINS-Mono only optimises frame states over a bounded-size sliding window, which causes a considerable drift in the trajectory.

In the pure monocular case, HFNet-SLAM has an average rms ATE of 2.8 cm, which outperforms ORB-SLAM3 and more than trebles the accuracy of DSM. However, due to fast motions, HFNet-SLAM fails to track in the *V203* sequence, as illustrated in Figure 5b. Due to the reliable loop detection module, this algorithm can still re-localise itself in previously visited locations and estimate parts of the trajectory with relatively low error in each execution. Compared with the pure vision case, the inertial integration enables the system to predict the camera motions when experiencing tracking loss, boosting the robustness and stability of the system.

To conclude, the appropriate application of deep features improves the scale estimation and tracking quality of the SLAM system. Therefore, HFNet-SLAM achieves more precise results than ORB-SLAM3 in both monocular and monocular-inertial cases.

#### 4.1.2. Visual-Inertial SLAM on TUM-VI Dataset

The circumstances in the EuRoC dataset are small, simple, and easy to solve. Therefore, the evaluation results in this benchmark may not represent its actual performance in real robot deployments. The TUM-VI dataset is more challenging than EuRoC dataset. It not only contains small indoor environments but also records environments with more difficult factors in the real-world scene, including corridors, outdoors, moving objects, and textureless structure [48]. However, TUM-VI dataset does not provide full ground-truth data for camera motions as the information is only captured in a room at the beginning and end of the trajectory. To avoid loop closure in these scenarios, viewpoint directions are opposite and no common region can be detected. Therefore, the evaluated ATE can still represent the accumulated drift during tracking.

During the evaluation, HFNet-SLAM takes 512 × 512 images as inputs. The system extracts 850 keypoints for every image to overcome the complexity of the environment. Figure 6 compares the accuracy of HFNet-SLAM using the monocular-inertial configuration with VINS-Mono and ORB-SLAM3. As the table shows, HFNet-SLAM has the most precise results among these systems in different scenarios. Especially in medium and large environments, HFNet-SLAM outperforms other systems in most of these sequences by a wide margin.

In small indoor environments, *room* and *corridor* sequences, the environment of these sequences is similar to that of the EuRoC dataset. Due to the advantages of region-reusing ability, both the HFNet-SLAM and ORB-SLAM3 systems achieve very good results in such environments. The average errors obtained by HFNet-SLAM are only 1 cm in *room* sequences and 8 cm in *corridor* sequences, which are smaller than other systems. These results further support the previous analysis in the EuRoC dataset evaluation that the employment of deep features brings benefits to the accuracy of the system in small indoor environments.

In medium and large environments, *magistrale* and *outdoors*, the performance of HFNet-SLAM is significantly better than other systems. In half of these sequences, HFNet-SLAM is twice as accurate as ORB-SLAM3 and has more than five times the precision of VIN-Mono. These are two key reasons for the improvement.

Firstly, the proposed loop detection strategy in HFNet-SLAM is accurate and reliable. In *magistrale* and *outdoors* sequences, common regions are rare and difficult to detect. Figure 7 records the performance of loop detection modules in these sequences where loops may exist. As can be seen, compared with ORB-SLAM3, HFNet-SLAM has higher recall and tends to recognise more correct loops in complex environments during the evaluation, resulting in up to a 97% reduction in trajectory errors. This is further explained in Figure 8, which compares trajectories estimated by HFNet-SLAM and ORB-SLAM3 in the *outdoors7* sequence. In this sequence, HFNet-SLAM successfully detects three potential loop closures during the tracking, while ORB-SLAM3 fails to take advantage of all loop information and misses two of them. The missed loop detection causes error accumulation in the trajectory of ORB-SLAM3, leading to an error of more than 7 m. The novelty of loop detection explains the dramatic improvement in accuracy.

Secondly, the high quality of local features in HFNet-SLAM leads to accurate pose estimation. Figure 9 and Figure 10 compare the performance of feature matching using ORB-SLAM3 and HFNet-SLAM in various environments during the evaluation. The matches with reprojection errors higher than 6 pixels are regarded as outliers. As can be seen, both algorithms have the highest matching precision in *room* sequences. However, as the environments get more complex, the precision significantly decreases in *magistrale* and *outdoors* sequences. In general, the tracking strategy of HFNet-SLAM tends to generate dense correspondences with high correctness in various environments. These results confirm the findings from both Sarlin et al. [16] and Tang et al. [32], which report that deep learning-based features have a higher matching score than hand-crafted algorithms. Excellent feature tracking provides sufficient and accurate pose estimation information and dramatically reduces the median ATE. This commonly occurs in some long sequences without loop closure. Despite errors in the order of 10–42 m in these sequences, HFNet-SLAM still has the best accuracy in large environments.

There are also three *slides* sequences that contain tubular slides with dim lighting. These sequences are extremely different for feature-based SLAM systems to solve. HFNet-SLAM lost tracking in such dart slides because of the insufficient number of features for extracting and tracking, but it can solve the whole sequence based on the IMU measurements. The HFNet-SLAM has a better estimation of the inertial parameters, including gravity direction, IMU bias, and scale, which results in small ATEs in these sequences. Due to tracking failures in dark environments, it is recommended to use the optical flow or direct method to process featureless images.

One considerable disadvantage appears during the evaluation. As shown in Figure 10, HFNet-SLAM failed to find any correct matches due to the large in-plane rotation. Such extreme in-plane rotation can be seen in *corridor* sequences and corrupts the accuracy of the SLAM system. The key reason for this problem is that the SuperPoint encoder in HF-Net is not rotationally invariant due to insufficient related training examples [9].

To conclude, the introduction of HF-Net boosts the accuracy of tracking and loop detection, reducing the median ATE error compared with other state-of-the-art systems. However, the used model, HF-Net, has bad performance when input images are featureless or have a large in-plane rotation, which has a negative impact on the overall performance.

### 4.2. Runtime Performance

Figure 11 records the mean and standard deviation of the time consumption of the main parts of HFNet-SLAM while evaluating the EuRoC dataset. As the table shows, HFNet-SLAM performs the main operations in tracking, mapping, and loop closure threads, which are the same as ORB-SLAM3.

In the tracking thread, both ORB-SLAM3 and HFNet-SLAM take 10 milliseconds for feature extraction, which means that the inference of the HF-Net model optimised by TensorRT has a similar computational cost as the traditional ORB algorithm. In the mapping thread, compared with ORB-SLAM3, despite fewer features being extracted for every frame, the keypoint matching strategy in HFNet-SLAM provides a greater number of correct feature matches. As a result of that, the mapping thread can provide a comparable number of map points and information, which leads to a similar running time and map complexity. In the loop closure thread, the proposed loop detection module in HFNet-SLAM is effective in filtering the wrong loop in the database query stage, avoiding unnecessary calculation in the next geometric verification. Therefore, it is simpler and more efficient than the BoW-based method in ORB-SLAM3.

In general, HFNet-SLAM is more effective than ORB-SLAM3 with the support of GPUs. This algorithm can run in real time at about 50 frames and 3–6 keyframes per second.

## 5. Conclusions and Future Work

Based on [1,16], this work proposes HFNet-SLAM, an accurate and real-time monocular SLAM system with deep features. The main contributions of this work, apart from the entire system itself, are the feature extraction strategy customized for deep CNNs as well as the keypoint matching and place recognition modules fully based on the generated local and global features. This work proves the practicality of incorporating deep features into the visual SLAM framework.

The experimental results showed that the appropriate application of deep features significantly improves the accuracy of pose estimation and loop detection and thereby boosts the accuracy of the entire HFNet-SLAM. The pure visual and visual-inertial configurations of HFNet-SLAM achieve the lowest errors, especially in complex and outdoor environments. Furthermore, with the support of GPU, SIMD, and TensorRT technology, HFNet-SLAM can run in real-time at 50 FPS.

There are two main disadvantages to HFNet-SLAM. The first one is the failure cases caused by the large in-plane rotation in camera motion because the used model, HF-Net, performs poorly with extreme rotation. Secondly, the calculation of this neural network needs the support of GPU accelerator resources, which may be impractical for mobile robot systems.

These drawbacks are the primary factors limiting the overall performance of the SLAM system. To address the problem, suggestions are given for future work. Firstly, when training neural networks for local feature extraction, it is reasonable to include more data with large in-plane rotation to improve rotation invariance. Secondly, some post-training optimisation tools, including quantization technology, can further reduce the cost of the HF-Net model, allowing the network to compute on CPUs. Last but not least, it is widely accepted that stereo and RGB-D SLAM have higher robustness and accuracy than the monocular configuration. Therefore, it is valuable to extend HFNet-SLAM to support different sensors.

## Figures and Tables

**Figure 1 sensors-23-02113-f001:**
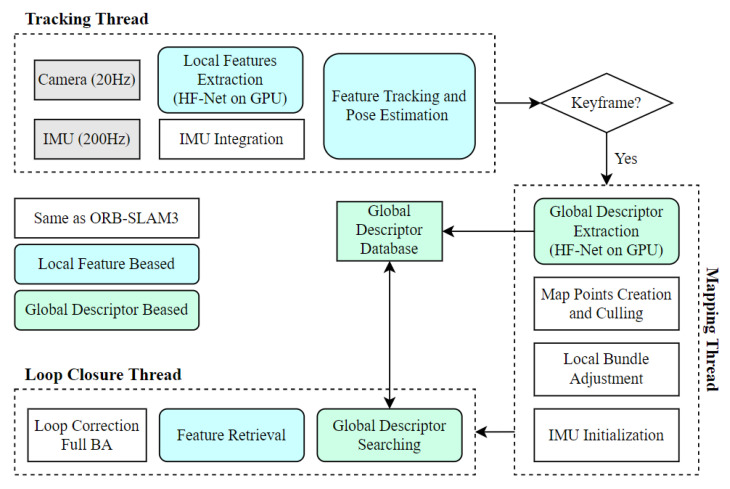
The main components of HFNet-SLAM framework.

**Figure 2 sensors-23-02113-f002:**
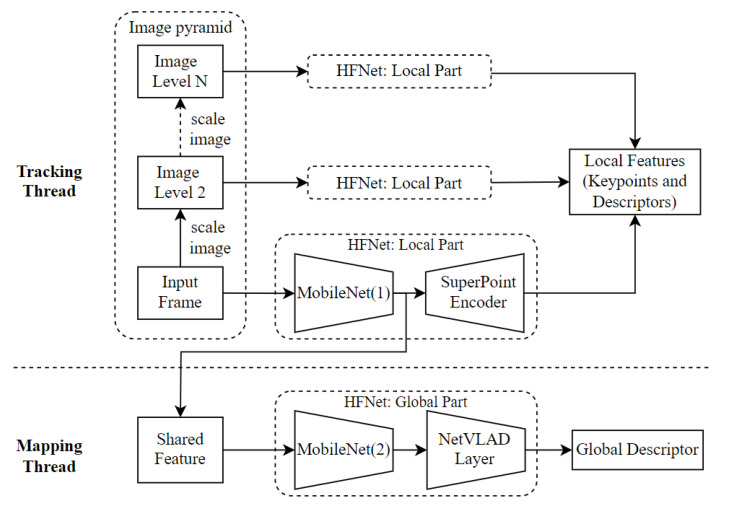
The feature extraction process of HFNet-SLAM.

**Figure 3 sensors-23-02113-f003:**
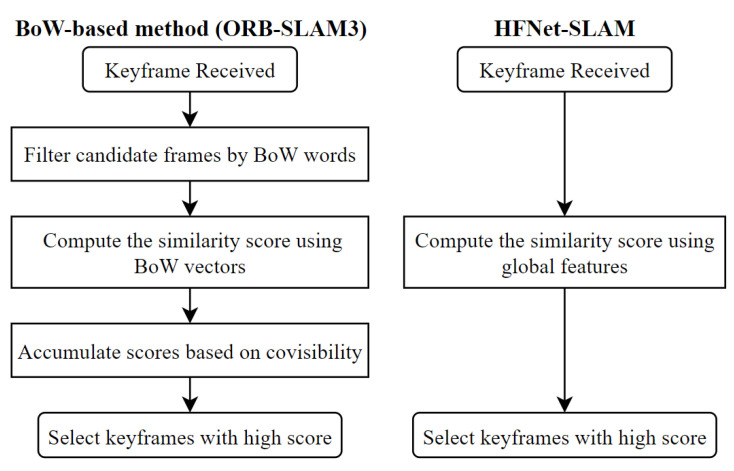
Plot of image retrieval modules in ORB-SLAM3 and HFNet-SLAM.

**Figure 4 sensors-23-02113-f004:**
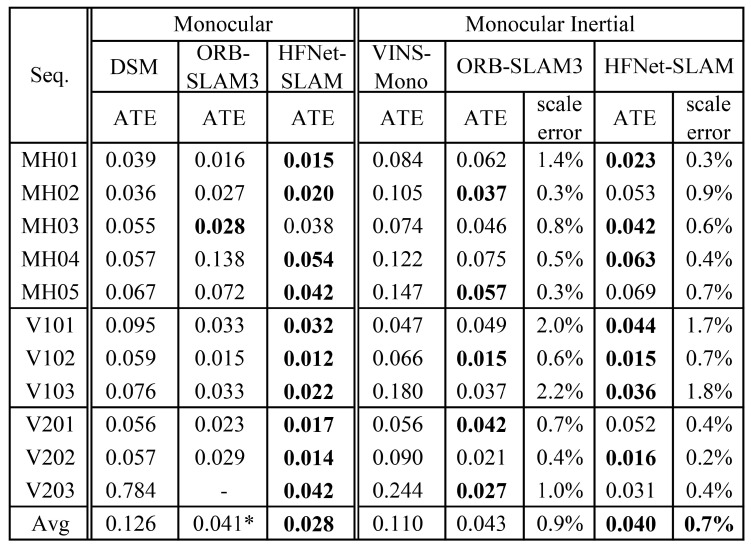
Comparison results in EuRoC dataset (rms ATE in meters; scale error in %). The results of HFNet-SLAM are median of ten executions. The measurements of other systems are provided by [1]. The bold values denote the smallest errors. Asterisks mark systems that did not complete all sequences.

**Figure 5 sensors-23-02113-f005:**
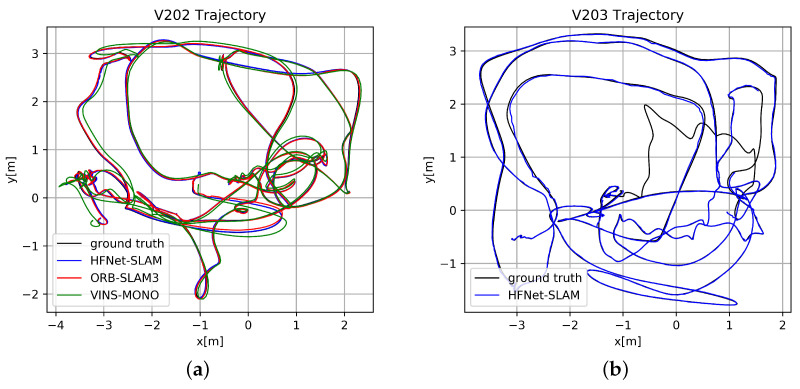
(**a**) Estimated trajectories in *V202* sequence with monocular-inertial configuration; (**b**) Estimated trajectories in *V203* sequence with pure monocular configuration.

**Figure 6 sensors-23-02113-f006:**
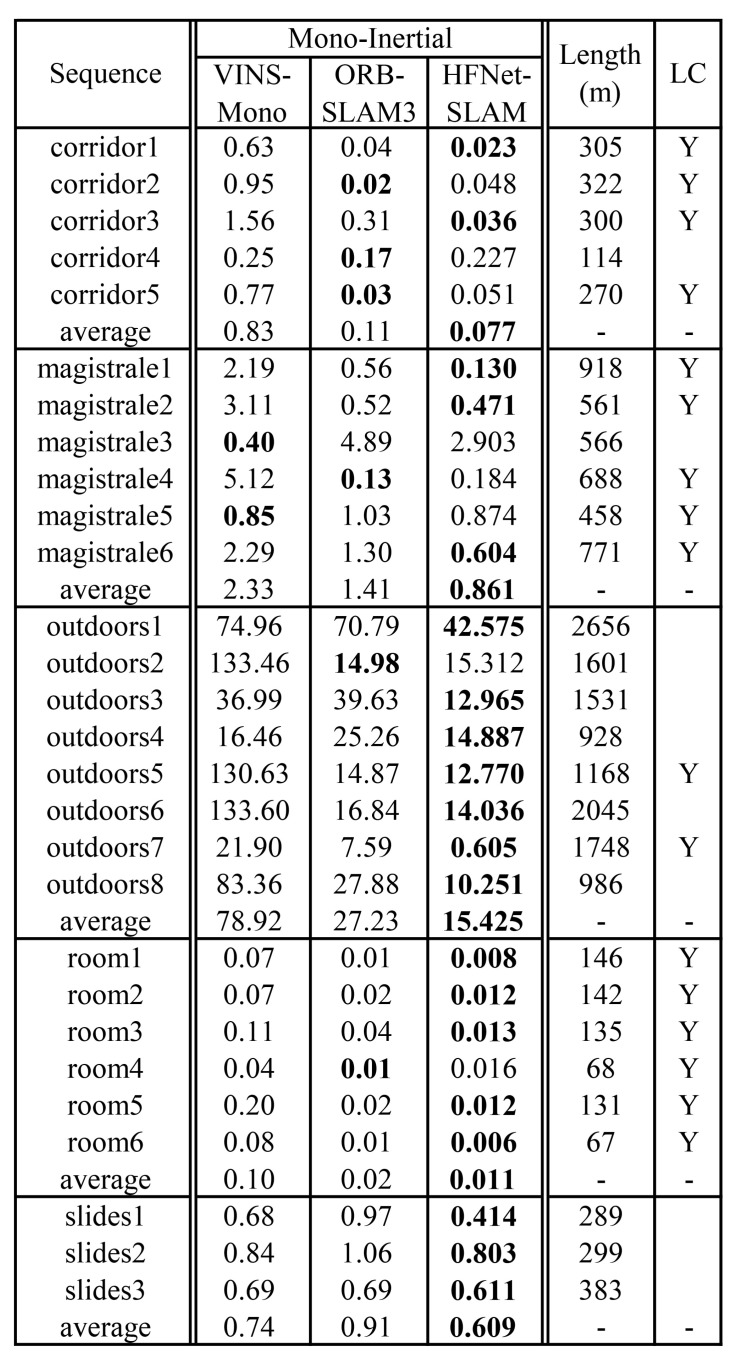
Comparisonresults in TUM-VI dataset (rms ATE in meters). The results of HFNet-SLAM are median of five executions. The measurements of other systems are provided by [1]. The bold values denote the smallest errors. LC means whether the loop closure may exist.

**Figure 7 sensors-23-02113-f007:**
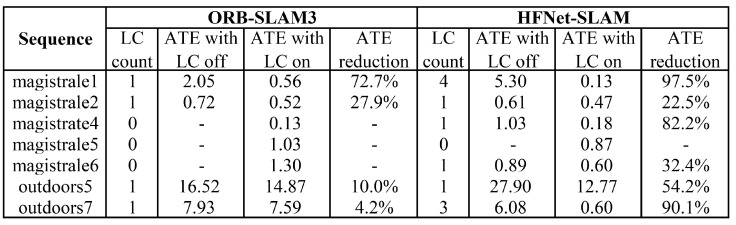
The comparison of loop detection modules in terms of the number of detected loops, the estimated ATE with and without loop closure, as well as corresponding error reduction (rms ATE in meters). The results are the median of five executions. The ATE errors with loop closure are provided by Figure 6.

**Figure 8 sensors-23-02113-f008:**
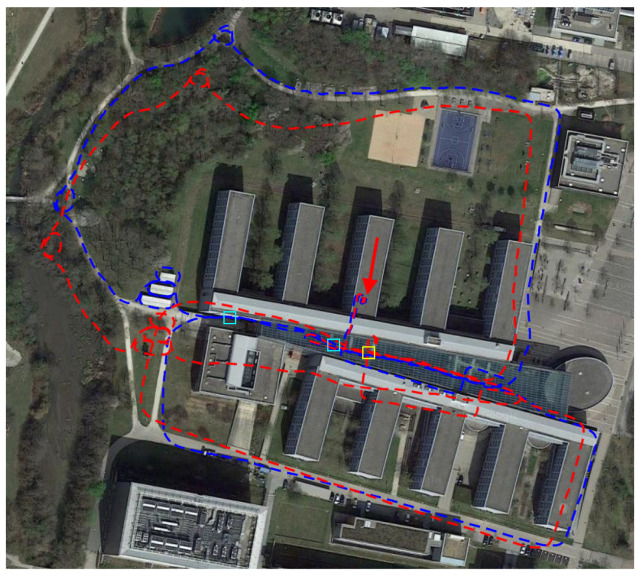
Estimated trajectories by HFNet-SLAM (blue) and ORB-SLAM3 (red) with the *outdoors7* sequence from TUM-VI dataset. Yellow squares denote the places for loop detection by both algorithms, while cyan squares denote the places for loop detection only by HFNet-SLAM. The red pointer points to the start and end of trajectories according to the ground truth data.

**Figure 9 sensors-23-02113-f009:**
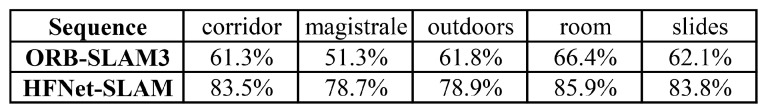
Theaverage precision of feature matching in various environments.

**Figure 10 sensors-23-02113-f010:**
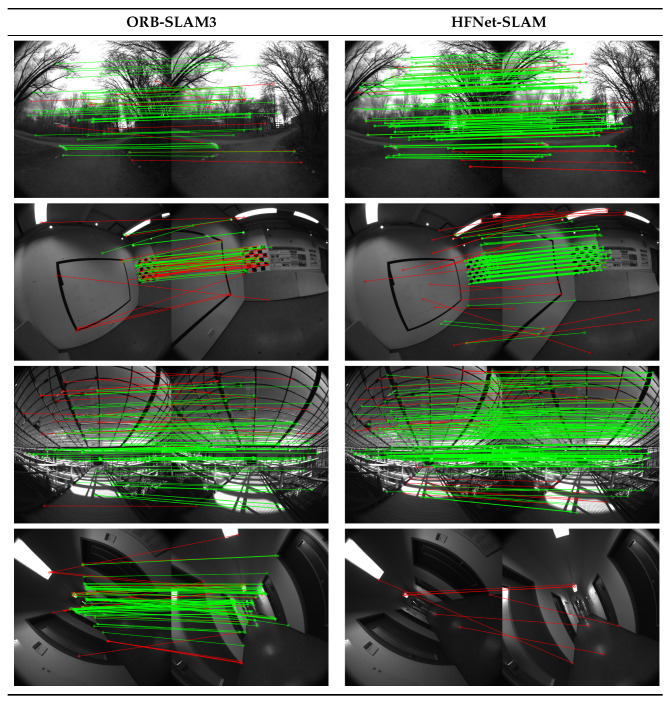
Qualitative results on feature matching. The green lines denote correct matches, while the red lines denote outliers. Row 4: Failure case due to extreme in-plane rotation.

**Figure 11 sensors-23-02113-f011:**
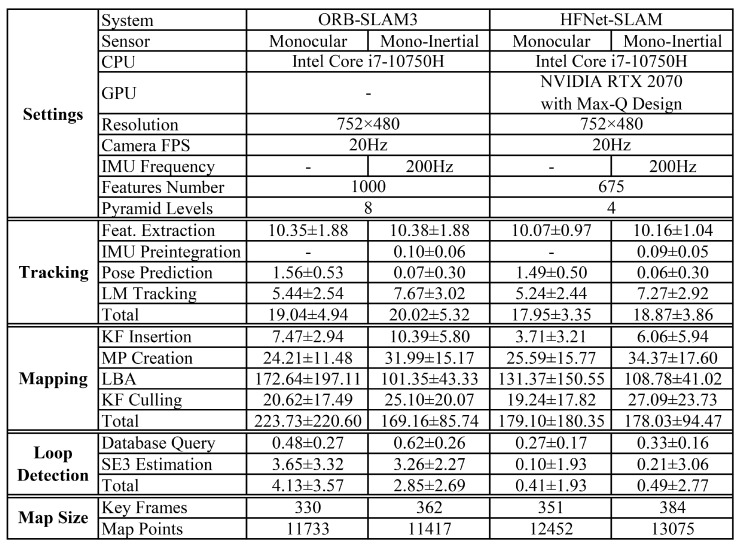
Processingtime of the main components of HFNet-SLAM compared with ORB-SLAM3, on EuRoC V202 (Time in millisecond).

## Data Availability

The source code is available at: https://github.com/LiuLimingCode/HFNet_SLAM (accessed on 5 February 2023).

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
