# Peer review of "HFNet-SLAM: An Accurate and Real-Time Monocular SLAM System with Deep Features"

_sensors, 2023, doi:10.3390/s23042113_

Round 1
Reviewer 1 Report
In this paper, the authors combine HF-net and ORB-SLAM3. A method for loop detection without BoW is provided, but there are still some problems.
Please consider the following comments:
1. As stated in the abstract, “The main novelty is the workflow of feature extraction, keypoint matching, and loop detection fully based on features from CNNs.”, while the authors make no changes to the original HF-net and some similar works, such as deepslam have been proposed, the authors should re-consider the novelty of the work.
2. DXSLAM can work without ROS. The difference between this work and DXSLAM should be re-considered.
3. The part of theory is too short and brief. The theory of the proposed method needs to be specific and improved.
4. The comparisons of trajectories on Euroc and TUM-VI dataset should be provided.
5. The authors extract 675 keypoints on Euroc dataset while extracting 850 keypoints on TUM-VI dataset. The authors should explain the reasons. The number of keypoints for ORB-SLAM3, DSM and VINS-Mono should also be provided.
6. DXSLAM use both BoW and global feature for loop detection. Is it better to use only global feature for loop detection?
7. Some self-collected datasets are needed to prove the effectiveness of the proposed method
Reviewer 2 Report
I appreciate the practical aspect of the work (real-time performance, tensorRT), but I have some doubts regarding the novelty of the presented work. I will list my doubts, starting with the most important ones:
1. I believe related work/background literature could be improved to match the current SoTA. I am the author of neither of these works, and I only wanted to point out the most trending ones.
A) Learnt matching:
SuperGlue: Learning Feature Matching With Graph Neural Networks
LoFTR: Detector-Free Local Feature Matching With Transformers
COTR: Correspondence Transformer for Matching Across Images
Why could these approaches not be applied in your solution? Would you expect future gains when not using HF-Net?
B) Fully trained SLAMs:
DROID-SLAM: Deep Visual SLAM for Monocular, Stereo, and RGB-D Cameras
NICE-SLAM: Neural Implicit Scalable Encoding for SLAM
Deep Patch Visual Odometry
We could agree/disagree on the future of SLAM, but comparing results with these works would be good.
C) Some similar work
RWT-SLAM: Robust Visual SLAM for Highly Weak-textured Environments
2. The choice of ORB-SLAM3 as a structural backbone leaves no doubts, but the choice of HF-Net should be convincing to the reader. Reading your paper, I felt this was the only option.
3. I understand that the strength of the approach lies in an ability to operate using monocular cameras but could you also show the results obtained with stereo ORB-SLAM3? My understanding of ORB-SLAM3's structure suggests that there should be no technical issues in presenting these results, which could improve the presentation.
4. I wish the presentation of the loop closure performance could be presented with more technical insight - the number of LCs for ORB-SLAM3 and your work for chosen sequence, the accuracy of LC based on GT, etc. It is hard to judge the real reason behind the improved performance.
5. Fig. 4 is ok, but I prefer more insight into comparing the number of features using ORB/HF-SLAM, the number (or ratio) of matched features, etc.
6. ``Recent development of deep convolutional neural networks (CNNs) has demonstrated strong ability in feature extraction [8].'' - are you sure that it is a proper citation for this statement?
7. ``Their models, called HF-Net`` - either model or HF-Nets
Overall, I believe improving related work and comparing your work to more systems is necessary for acceptace.
Reviewer 3 Report
There are no comments.
Author Response
We thank the reviewer for their work. There are no comments requiring reply.
Round 2
Reviewer 1 Report
The concerns have been addressed.
Author Response
We thank Reviewer 1 for their comments and for being positive about the paper.
Reviewer 2 Report
Thank you for the reply.
I understand limited resources to extend this work, but more insight is expected out of the journal submission:
A) The evaluation on TUM-VI requires LC to be turned off, as GT is only available at the beginning and at the end. Having LC in these scenarios warps the results as the system has to be accurate at the start and have a well-working LC at the end.
B) The presented average precision:
- requires formal definition, e.g., to clarify what reprojection error threshold was used to determine whether a match is correct or incorrect
- table 3 suggests that ORB-SLAM3 matching outperforms HFNet-SLAM, so why does HFNet obtain better accuracy? Is it due to the greater number of sensible features? This has to be explained to the reader and provide a more in-depth understanding
C) We still lack any evaluation of the LC to state that the proposed approach outperforms BoW-based LC
Round 3
Reviewer 2 Report
Thanks for all the corrections.
I would recommend making minor adjustments to table 3:
- to also present the numerical value with LC (apart from the percent reduction) as it requires the reader to go back to table 2 to get raw values
- change 'Detected LC' to 'No. of LCs` or 'LC count.'
Author Response
We thank you for your work reviewing the article and providing comments that have led to all the improvements.
- to also present the numerical value with LC (apart from the percent reduction) as it requires the reader to go back to table 2 to get raw values
We have added this column to the table and clearly indicated it.
- change 'Detected LC' to 'No. of LCs` or 'LC count.'
We have changed the column header to LC Count.